# Intestinal Ischemia/Reperfusion Injury Influences Hyaluronan Homeostasis in the Rat Brain

**DOI:** 10.3390/ijms262010064

**Published:** 2025-10-16

**Authors:** Annalisa Bosi, Nicolò Baranzini, Alessandra Ponti, Paola Moretto, Elisabetta Moro, Francesca Crema, Rossella Cianci, Evgenia Karousou, Manuela Viola, Alberto Passi, Davide Vigetti, Andreina Baj, Annalisa Grimaldi, Paolo Severgnini, Cristina Giaroni

**Affiliations:** 1Department of Medicine and Technological Innovation, University of Insubria, 21100 Varese, Italy; aponti@uninsubria.it (A.P.); andreina.baj@uninsubria.it (A.B.); cristina.giaroni@uninsubria.it (C.G.); 2Department of Biotechnology and Life Sciences, University of Insubria, 21100 Varese, Italy; nicolo.baranzini@uninsubria.it (N.B.); annalisa.grimaldi@uninsubria.it (A.G.); paolo.severgnini@uninsubria.it (P.S.); 3Department of Medicine and Surgery, University of Insubria, 21100 Varese, Italy; paola.moretto@uninsubria.it (P.M.); jenny.karousou@uninsubria.it (E.K.); manuela.viola@uninsubria.it (M.V.); alberto.passi@uninsubria.it (A.P.); davide.vigetti@uninsubria.it (D.V.); 4Department of Internal Medicine and Therapeutics, University of Pavia, 27100 Pavia, Italy; elisabetta.moro@unipv.it (E.M.); francesca.crema@unipv.it (F.C.); 5Department of Translational Medicine and Surgery, Catholic University of the Sacred Heart, 00168 Rome, Italy; rossella.cianci@unicatt.it; 6Fondazione Policlinico Universitario A. Gemelli, Istituto di Ricovero e Cura a Carattere Scientifico (IRCCS), Largo Agostino Gemelli 8, 00168 Rome, Italy; 7Neuroscience Center, University of Insubria, 21100 Varese, Italy

**Keywords:** intestinal ischemia/reperfusion injury, brain, extracellular matrix, hyaluronan, 4-methylumbelliferone, inflammation

## Abstract

Intestinal ischemia and reperfusion injury (IRI) can lead to multiple organ dysfunction, including the central nervous system (CNS), where a neuroinflammatory response may develop. Hyaluronan, a glycosaminoglycan component of the extracellular matrix, has been shown to modulate enteric neuronal and immune function during in vivo IRI in the rat small intestine. The aim of this study was to investigate the potential involvement of hyaluronan in the alterations induced by in vivo intestinal IRI in the rat hippocampus and striatum. Mesenteric ischemia was induced in anesthetized adult male rats for 60 min, followed by 24 h of reperfusion. Injured (IRI group), sham-operated (SHAM group), and non-injured (CTR group) animals were treated with the hyaluronan synthesis inhibitor 4-methylumbelliferone (4-MU; 25 mg/kg). In the hippocampus and striatum of the IRI group, levels of both hyaluronan and neurocan, a proteoglycan primarily found in the central nervous system extracellular matrix, as well as the hyaluronan synthesizing enzyme Has2, were significantly downregulated compared to the CTR and SHAM groups. These changes were associated with alterations in the TLR4-NFκB-pIκB pathway, with the effects being more prominent in the hippocampus than in the striatum. Increased levels of IL6, co-localizing with the microglial marker S100β, were observed in both regions and were attenuated by 4-MU only in the hippocampus. Overall, these findings suggest that intestinal IRI may disrupt extracellular matrix homeostasis and induce hyaluronan-mediated enhancement of local proinflammatory signaling, primarily involving IL6 and microglial cells, mainly in the hippocampus. Such changes may contribute to the development of cognitive deficits and memory dysfunction associated with intestinal IRI.

## 1. Introduction

Intestinal ischemia–reperfusion injury (IRI) is a life-threatening clinical condition that can result from mesenteric artery embolism, intestinal obstruction, sepsis, shock, trauma, and surgery [1]. The initial ischemic insult induces mucosal shedding, barrier dysfunction, and bacterial translocation, accompanied by a prolonged reduction in intestinal blood flow. The subsequent reperfusion phase is critical for rescuing ischemic tissue; however, it also exacerbates cellular injury and vascular and tissue damage [1,2,3]. Intestinal IRI has severe consequences for all the cell types composing the enteric microenvironment, including epithelial and smooth muscle cells, glial cells, and neurons [3,4]. The local production of proinflammatory cytokines, such as Tumor Necrosis Factor-*α* (TNF-*α*), Interleukin 6 (IL6), and Interleukin 1β (IL1β), as well as reactive oxygen species (ROS) and nitric oxide (NO), and the increased expression of vascular and intercellular adhesion molecules (such as ICAM-1), are well-established processes that undermine the enteric microenvironment during intestinal IRI [3,4,5,6]. It is now well acknowledged that these inflammatory signaling pathways may also affect distant sites, including the brain, leading to increased blood–brain barrier permeability, permeation of proinflammatory circulating factors, and activation of microglia [7,8,9]. In rat models of intestinal IRI, microglial activation was associated with persistent neuroinflammation and cognitive impairment, despite intestinal recovery [8,9]. These manifestations were linked to alterations in brain levels of proinflammatory factors, increased oxidative stress, apoptosis, and elevated levels of proteins associated with inflammation, including Toll-like receptor 4 (TLR4), the downstream myeloid differentiation factor 88 (MyD88), and phosphorylated nuclear factor kappa B (NF-κB) [9].

Hyaluronan is a ubiquitous extracellular matrix glycosaminoglycan formed by alternating glucuronic acid and N-acetylglucosamine disaccharides by three isoenzymes named hyaluronan synthase 1, 2, and 3 (HAS1, 2, and 3). In the central nervous system (CNS), hyaluronan is widely distributed throughout white matter and also highly concentrated in gray matter, including perineuronal nets [10]. HAS1, HAS2, and HAS3 are located on the plasma membrane of various cell types in the CNS, including glial cells, astrocytes, and more specifically in neurons, exhibiting varying molecular weights and synthesis rates that depend on the specific region of the CNS and developmental stages [11,12,13,14]. Additionally, HAS enzymes are found within the cell, where nascent forms follow the secretory pathway to the membrane, or are recycled from the membrane itself [15,16]. The specific role of each HAS enzyme is still under investigation, but several critical functions of hyaluronan are regulated by HAS2 which is probably the most active enzyme in the body [14]. In the nervous system, hyaluronan, along with chondroitin sulfate proteoglycans (e.g., aggrecan, brevican, neurocan) and link proteins such as the brain-specific link protein (Bral1 and Bral2) and hyaluronan and proteoglycan link protein 1 (HAPLN1), contributes to the formation of perineuronal nets, which are vital for neuronal communication, synaptic function, protein mobility, and protection against oxidative stress [13,17,18]. Additionally, the soft extracellular matrix in the brain, rich in glycosaminoglycans and hyaluronan-associated proteins, is involved in the regulation of cell adhesion, migration, and neurite outgrowth [19,20]. Hyaluronan contributes to the development of the IRI-associated proinflammatory state in the gut by promoting harmful bacterial overgrowth, enhancing neutrophil infiltration, and causing alterations in the architecture and function of myenteric neurons [21,22]. In the enteric nervous system (ENS), alike in the central nervous system, hyaluronan and its interacting molecules contributes to the external architecture of enteric ganglia and also participates in the formation of a perineuronal net surrounding myenteric neurons [21,22,23]. The biological effects of hyaluronan are mediated by a wide range of different hyaluronan cell receptors including cluster differentiation 44 (CD44), receptor for hyaluronan-mediated motility (RHAMM), lymphatic vessel endothelial hyaluronan receptor 1 (LYVE-1), and hyaluronan receptor for endocytosis (HARE). Among these, CD44 is one of the most extensively studied and represents the main hyaluronan receptor expressed in tissues. Furthermore, although TLRs have been extensively studied as pathogen-associated molecular pattern receptors involved in innate immunity, both TLR2 and TLR4 also participate in hyaluronan signaling involving hyaluronan fragments [24]. Interestingly, in the rat gut, IRI-induced enhancement of endogenous hyaluronan levels is involved in the upregulation of TLR2 and TLR4 in the muscularis propria and submucosal layers.

In this context, the aim of this study was to test the hypothesis that intestinal IRI may induce changes in endogenous hyaluronan homeostasis and related proinflammatory pathways in the brain. To this end, we evaluated the effects of temporary occlusion of the superior mesenteric artery in male rats, focusing on the striatum and hippocampus in both the absence and presence of 4-methylumbelliferone (4-MU), which inhibits hyaluronan production in various tissues and cell lines by reducing the cytosolic precursor UDP-glucuronic acid and potentially downregulating the transcription of HAS2 [25,26].

## 2. Results

### 2.1. Intestinal IRI Influences Hyaluronan Homeostasis and Perineuronal Net Molecular Components in the Rat Brain

Hyaluronan levels in the striatum and hippocampus were evaluated by using hyaluronan binding protein (HABP). In the hippocampus, HABP stained both neurons and glial cells, as evidenced by the presence of both cytoplasmic and extracellular signal visible in the intercellular region. HABP staining was particularly intense in the CA1 region as shown in Figure 1A,B. Both in the IRI and SHAM groups, HABP staining was significantly reduced with respect to the CTR group, and such reduction was unaffected by 4-MU. In the IRI group, HABP staining was significantly lower than in the SHAM group (Figure 1B). In the IRI and SHAM groups, changes in hyaluronan deposition were associated with a significant reduction in hyaluronan synthase 2 (*Has2*) mRNA levels, while *Has1* and *Has3* transcript levels did not change in all experimental conditions compared to CTR values (Figure 2A–C). 4-MU treatment induced a significant reduction in *Has2* mRNA levels in CTR, SHAM, and intestinal IRI groups (Figure 2B). No differences in the mRNA levels of hyaluronidase (*Hyal*) *1,2* and *3*, the endoglycosidase enzymes that break down hyaluronan, were observed among all the experimental groups (Figure 2D–F). We next evaluated the transcript expression of two of the major chondroitin sulfate proteoglycan components of the perineuronal nets, aggrecan (*Acan*) and neurocan (*Ncan*). In the hippocampus of the different experimental groups, *Acan* mRNA levels were not significantly different with respect to CTR values (Figure 3A), while *Ncan* transcript levels showed a trend towards a reduction, which reached significance only in the IRI-4MU group (Figure 3B). NCAN staining in the CA1 region of the hippocampus was observed especially in neurons (Figure 3C,D). In agreement with qRT PCR data, NCAN fluorescence intensity was significantly reduced in the SHAM and IRI groups with respect to CTR values. In all experimental groups, NCAN fluorescence further decreases in the presence of 4-MU (Figure 3C).

In the striatum, HABP staining was observed in both neurons and glial cells, with a prominent signal that appeared more dispersed within the extracellular matrix (Figure 4A). As observed in the hippocampus, in the striatum of the IRI and SHAM groups the intensity of HABP fluorescence was significantly reduced with respect to CTR values and was unaffected by 4-MU treatment (Figure 4A,B). In the striatum of the IRI group a significant reduction in *Has2* mRNA levels was observed with respect to CTR values, while *Has1* and *Has3* transcript levels did not change significantly in all the experimental conditions (Figure 5A–C). In all experimental groups, 4-MU treatment induced a significant reduction in HAS2 transcript levels with respect to non-treated CTR values (Figure 5B). In all the experimental groups, the expression of *Hyal1*, *Hyal2*, and *Hyal3* mRNA did not differ significantly (Figure 5D–F).

Striatal *Acan* mRNA levels were also similar to CTR values in all experimental groups, while *Ncan* transcript levels were significantly reduced in the IRI and IRI-4MU groups with respect to CTR values (Figure 6A,B). In the SHAM group, a trend towards a reduction in *Ncan* mRNA levels was observed which reached significance after 4-MU treatment (Figure 6B). Analogously to the hippocampus CA1 region, in the striatum, NCAN immunoreactivity was found in neurons (Figure 6C,D). NCAN fluorescent intensity was significantly reduced in SHAM and IRI groups, both in the absence and presence of 4-MU with respect to the CTR group (Figure 6C).

### 2.2. Effect of 4-MU on Hyaluronan Receptor Expression in the Rat Hippocampus and Striatum After Intestinal IRI

We next assessed the mRNA and protein levels of the two main hyaluronan receptors: TLR4, which is modulated by hyaluronan during inflammatory responses, and CD44, the most abundant hyaluronan receptor in the brain [27]. In the hippocampus, both *Tlr4* mRNA and its protein levels were reduced in the IRI and SHAM groups compared to CTR values, and this decrease reached significance in the presence of 4-MU (Figure 7A,C). 4-MU treatment also induced a significant reduction in *Tlr4* expression in the CTR group (Figure 7A,C). In contrast, *Cd44* transcript and its protein levels were comparable across all experimental groups (Figure 7B,D). In the striatum, both *Tlr4* and *Cd44* mRNA and their protein levels remained unchanged across groups (Figure 7E–H).

To investigate potential intracellular pathways downstream of hyaluronan receptors, we measured *Nfkb* mRNA expression and the protein levels of the phosphorylated form of IκBα (pIκBα), a key regulator of the NF-*κ*B signaling cascade, in both brain regions. Indeed, the phosphorylation of the inhibitor of κB (IKB) proteins by IκB Kinase β (IKKβ) leads to IκB degradation and subsequent activation of nuclear factor κB [28]. In the hippocampus, *Nfkb* mRNA levels were significantly reduced in the IRI and SHAM groups compared to CTR values and remained downregulated after 4-MU treatment (Figure 8A). Protein levels of pIκBα were significantly increased in the IRI group compared to CTR and SHAM but returned to CTR values after 4-MU treatment (Figure 8B). In the SHAM group, pIκBα protein levels were similar to CTR values and 4-MU had no effect on either SHAM or CTR animals (Figure 8B). In the striatum, *Nfkb* expression did not differ between groups (Figure 8C). However, pIκBα protein levels were significantly increased in the SHAM group compared to CTR, an effect reversed by 4-MU (Figure 8D). In the IRI group, pIκBα levels were similar to those observed in the CTR group and were unaffected by 4-MU treatment (Figure 8D).

### 2.3. Effect of 4-MU Treatment on Intestinal IRI-Induced Changes in Proinflammatory Cytokines in the Rat Brain

In both the hippocampus and striatum, *Il6* mRNA levels were significantly increased compared to CTR animals (Figure 9A,C). This enhancement was not observed in the striatum and hippocampus of the SHAM group, suggesting that the increase is specifically associated with IRI. In both brain regions, 4-MU treatment significantly reduced IRI-induced *Il6* mRNA levels; however, in the striatum, these levels remained significantly higher than those in the CTR group (Figure 9A,C). In the hippocampus, *Tnfa* mRNA levels were similar in the IRI and SHAM compared to the CTR group (Figure 9B). In all experimental groups, 4-MU treatment significantly reduced hippocampal *Tnfa* mRNA levels (Figure 9B). In the striatum, *Tnfa* mRNA levels were similar across all experimental groups (Figure 9D).

### 2.4. 4-MU Modulation of Microglia in the Rat Brain After Intestinal IRI

In the CNS, proinflammatory cytokines are produced by astrocytes, microglia, and neurons in response to diverse intrinsic and extrinsic stimuli [29]. We, therefore, evaluated IL6 distribution by immunohistochemistry in both the hippocampus and striatum. In both regions, IL6 staining was localized to glial cells, showing a clear cytoplasmic signal (Figure 10A and Figure 11A). Consistent with q-RT-PCR data, IRI significantly increased IL6 immunoreactivity (IR) in the CA1 hippocampal region and in the striatum, and this effect was reduced by 4-MU treatment (Figure 10A,B and Figure 11A,B). In the hippocampus of the SHAM group, IL6 IR was also increased despite unaltered *Il6* mRNA levels, although the enhancement was less pronounced than in the IRI group. In both regions, IL6 staining colocalized with S100β, a member of the multigenic family of Ca^2+^-binding proteins abundantly expressed in astrocytes and microglial cells [30]. The number of IL6^+^/S100β^+^ cells increased in both the CA1 region and in the striatum after IRI, suggesting the possible involvement of microglia in the CA1 region (Figure 10A,B and Figure 11A,B). Interestingly, S100β expression followed different trends in the two regions. In the hippocampal CA1 region, both S100β fluorescence intensity and its mRNA levels were similar across CTR, SHAM, and IRI groups (Figure 10C,D). However, 4-MU significantly reduced S100β IR and its transcript levels in all groups (Figure 10C,D). Conversely, in the striatum, S100β fluorescence intensity was increased in the IRI group compared to the CTR and SHAM, and this effect was not altered by 4-MU treatment (Figure 11C). In the striatum, *S100β* transcript levels were similar in all experimental groups (Figure 11D).

## 3. Discussion

Intestinal IRI is a severe clinical condition favoring the development of systemic inflammation, which may extend to the brain. Although the correlation among IRI, neuroinflammation, and cognitive impairment has been assessed in several studies, the underlaying molecular mechanisms are still largely unknown [7,8,9]. In this study, we report the possible involvement of hyaluronic acid in the rat hippocampus and striatum following in vivo IRI in the small intestine.

Hyaluronan is the most abundant matrix polysaccharide in the adult brain and plays crucial roles in modulating neuron and glial cell differentiation, neuron activity, and repair in neurodegenerative diseases and after CNS injuries [17]. In this study, in both the rat hippocampus and striatum, hyaluronan binding protein stained the cytoplasm of both neurons and glial cells, consistent with the well-established evidence that neurons, astrocytes, and microglia are cellular sources of hyaluronan in the brain [17]. Furthermore, a more diffuse extracellular staining was evident in both regions, consistent with the role of hyaluronan as an extracellular matrix component. Following intestinal injury, hyaluronan deposition was significantly reduced in both the CA1 region of the hippocampus and the striatum. In agreement with data obtained in the mouse hippocampus, changes in hyaluronan deposition after intestinal IRI were observed in the CA1 region of the hippocampus, where a deficit of hyaluronan was suggested to underlie alterations in neuronal activity and epilepsy [31]. The decrease in hyaluronan levels in the rat brain after IRI may depend either on a downregulation of hyaluronan synthases or on an enhanced activity of the hyaluronan-degrading enzyme hyaluronidase (HYAL) [19,20]. In good agreement with the decrease in hyaluronan deposition, *Has2* mRNA levels decreased in both the hippocampus and striatum of the IRI group. The extent of *Has2* mRNA reduction was higher in the CA1 region of the hippocampus than in the striatum, in line with the more obvious reduction of hyaluronan binding protein staining in this region. It is highly probable that HAS2, the main biologically active isoform of HAS enzymes, rather than HAS1 and HAS3, participates in the reduction in hyaluronan deposition in the rat brain following intestinal IRI in both brain regions, since *Has1* and *Has3* mRNA levels remained unchanged in this condition. Both in the hippocampus and in the striatum of all experimental groups, 4-MU significantly decreased *Has2* mRNA levels and did not influence *Has1* and *Has3* transcript levels, suggesting a higher sensitivity of *Has2* to 4-MU modulation as observed in the rat enteric nervous system and in rat astrocytes [21,22,32]. However, in both regions, hyaluronan deposition was not influenced by 4-MU after IRI, possibly due to compensatory activity by other HAS isoforms that may have substituted for HAS2 to sustain hyaluronan production. HAS3, which rapidly produces hyaluronan, even with a minimal amount of substrate [33], may be involved since this isoform plays a fundamental role in the cell packing of the stratum pyramidal in the CA region of the hippocampus [31]. 

Hyaluronan signaling in the brain is strictly dependent on its molecular weight. While native long hyaluronan polymers, with high molecular weight, can influence a variety of biological processes and reduce the development of pathological states, including neuroinflammation, their cleavage into smaller fragments may promote immune cell activation as well as the production of proinflammatory cytokines, thereby favoring an increased inflammatory response [34]. However, in our study, mRNA levels of the three major hyaluronan-degrading enzymes (*Hyal1*, *Hyal2*, and *Hyal3*) did not shown significant differences both in the hippocampus and striatum, suggesting that alteration of hyaluronan deposition mainly depends upon its synthesis. 

In both the hippocampus and striatum of the SHAM group, hyaluronan staining was significantly lower than in CTR animals, suggesting that surgical manipulation and laparotomy may reduce hyaluronan deposition in the brain. The reduction in hyaluronan deposition after sham operation was, however, significantly lower than in the IRI group, indicating that IRI, per se, decreased hyaluronan levels. 

As regards the different proteoglycans contributing to the composition of the perineuronal net, immunoreactivity for neurocan, but not for aggrecan, was significantly reduced in the rat brain following intestinal IRI, with a more pronounced reduction in the CA1 region of the hippocampus. The molecular mechanism(s) underlying this change appears to differ between the two regions, as *Ncan* mRNA levels were only slightly reduced after intestinal IRI, suggesting that post-transcriptional and/or degradation processes may be involved. In contrast, a significant reduction in the *Ncan* transcript was observed in the striatum. Differences in the mechanisms underlying neurocan deposition in the two regions are also reflected by the sensitivity of hippocampal neurocan-IR to 4-MU, which was not observed in the striatum in all experimental groups. In this context, a recent study has shown that chronic oral treatment with 4-MU may reduce the perineuronal nets formation in the mouse hippocampus by diminishing both hyaluronan and neurocan levels, thus influencing memory retention [35].

Considering the downstream mechanisms of hyaluronan cell signaling, CD44, which plays various roles in both the developing and adult nervous system, appeared unaffected by intestinal IRI, as its mRNA levels remained unchanged in both regions [17]. However, in the hippocampus of the IRI group, but not in the striatum, downregulation of TLR4 mRNA levels corresponded well with the reduced hyaluronan deposition. 4-MU significantly decreased TLR4 mRNA under control conditions, but not after sham operation or IRI, suggesting that the inhibitory effect may act differently depending on tissue conditions. The transcript levels of *Nfkb*, a transcription factor downstream to TLR4, were reduced only in the hippocampus following intestinal IRI. This reduction was compensated by increased protein levels of the phosphorylated form of NF-κB inhibitory protein IκBα, promoting NF-κB activation [36]. Intestinal IRI-induced pIκBα protein levels in the hippocampus were significantly reduced by 4-MU, suggesting that hyaluronan may positively modulate the multi-subunit IκB kinase (IKK), as previously observed in prostate cancer cells, possibly via hyaluronan receptors alternative to TLR4 [37]. In the CNS, activation of NF-κB, principally occurring in activated glial cells, controls the induction of proinflammatory cytokines and chemokines, sustaining neuroinflammation [38]. Accordingly, in the hippocampus of the IRI, but not of the sham group, *Il6* mRNA levels and immunoreactivity significantly increased, indicating an IRI-specific effect on the cytokine transcription. In the striatum of the IRI group, although neither *Nfkb* transcription nor IκBα were modified with respect to control animals, *Il6* transcript levels and IL6-IR were significantly enhanced, suggesting alternative mechanisms underlaying the cytokine upregulation. Interestingly, in both regions, 4-MU significantly reduced *Il6* transcript after IRI, suggesting the involvement of hyaluronan. However, we cannot exclude alternative mechanisms since 4-MU may also exert anti-inflammatory effects in a hyaluronan-independent manner, for instance by modulating STAT3 or other signaling pathways [32,39]. pIκBα levels increased, indicating functional activation of the NF-κB pathway. This apparent discrepancy can be explained by the post-transductional regulation of TLR4, as well as by the contribution of alternative hyaluronan receptors. The reversal of pIκBα and IL6 induction by 4-MU further supports the functional link between hyaluronan metabolism and NF-κB activation. Overall, although pIκBα analysis alone does not fully reflect NF-κB activity, our data integrating hyaluronan deposition, TLR4, and *Nfkb* transcript levels, pIκBα protein, and IL6 induction provide a coherent picture supporting hyaluronan–TLR4–NF-κB signaling in the hippocampus after intestinal IRI. The consistent attenuation by 4-MU further supports the involvement of this pathway. Future studies, including NF-κB nuclear translocation or receptor blockade, would strengthen this mechanistic link.

In contrast to a previous study showing that *Tnfa* mRNA levels increased in the rat hippocampus and remained unchanged in the striatum after intestinal IRI [7], in our model, the levels of the cytokine remained unchanged in both regions. This discrepancy may be due to differences in our IRI model, which involved 60 min of ischemia and 24 h of reperfusion, potentially allowing cytokine levels to return to normal, as compared to the 30 min of ischemia and 6 h of reperfusion used in the previous study. Several reports suggest that microglia play an important role in the production of proinflammatory cytokines in the brain [40]. Accordingly, in this study, double fluorescence staining in the hippocampus and striatum of the IRI group revealed increased IL6/S100*β* co-staining in microglial cells, indicating the involvement of microglia in sustaining cytokine upregulation, in good agreement with previous studies [7,8,9]. However, S100*β*-immunoreactivity (S100*β*-IR) increased only in the striatum and remained unchanged in the hippocampus, suggesting that different mechanisms may underlie glial activation in the two regions [29]. This hypothesis was further supported by the sensitivity of S100*β*-IR and mRNA to 4-MU in the hippocampus but not in the striatum, indicating potential regional differences in hyaluronan-mediated modulation of microglia [41]. In this context, we cannot exclude that proinflammatory mediators released by the gut after intestinal IRI may be transferred to the brain, inducing cytokine synthesis in the two brain regions, thereby influencing hyaluronan deposition [42]. Indeed, the existence of a bidirectional gut–brain communication axis, which may contribute to the development of both local and CNS symptoms associated with gut diseases, is now widely acknowledged [43].

Overall, our data suggest that hyaluronan contributes to intestinal IRI-induced damage in the rat hippocampus, more so than in the striatum. Differences in ECM composition and structure, cell types, architecture, and plasticity may explain such a discrepancy. Indeed, since the hippocampus is characterized by a rich laminar ECM, more dynamic PNN structure, and higher plasticity than the striatum, in this region hyaluronan machinery may be more susceptible to injury [44]. Alteration of extracellular matrix components by neuroinflammatory conditions has been suggested to favor the development of neurological and neurodevelopmental disorders. From a translational viewpoint, our model may mimic a mild intestinal ischemic condition, which occurs, for example, during surgical extracorporeal circulation, which is associated with transient cognitive deficits after arousal [45,46]. Since rat tissues have reduced hyaluronan metabolism, we may expect that the abundance of catabolized hyaluronan fragments may be higher in human brain regions after intestinal IRI, further strengthening the development of neuroinflammation [47].

In summary, our data show alterations in hyaluronan homeostasis in the rat brain following intestinal IRI, with more significant consequences in the hippocampus. We cannot exclude that proinflammatory mediators, released by the gut after intestinal IRI, may be transferred to the brain, influencing hyaluronan hyaluronan deposition. A more in-depth assessment of the various mechanisms described in the present study is needed to gain a clearer understanding of the role of hyaluronan in the brain after intestinal IRI, potentially paving the way for new therapeutic strategies.

## 4. Materials and Methods

### 4.1. Animals

Male Wistar rats (weight 250–350 g, Envigo, Udine, Italy) were housed under controlled environmental conditions (temperature 22 ± 2 °C; relative humidity 60–70%) with a regular 12/12 h light/dark cycle and free access to standard laboratory chow and tap water ad libitum at the Animal Facility of the University of Insubria. 

Rats were assigned into six groups (N = 5 per group): (i) the sham-operated animals (SHAM), undergoing exposure of superior mesenteric artery (SMA), without occlusion, (ii) the intestinal IRI group (IRI), where the SMA was isolated by laparotomy, and a non-invasive clamp was released 60 min after clamping as described in the next paragraph, (iii) normal un-operated rats, which were used as controls (CTR), (iv) 4-methylumbelliferone (4-MU)-treated sham group (SHAM-4MU), (v) 4-MU-treated IRI group (IRI-4MU), and (vi) normal-4MU-treated group (CTR-4-MU). 4-MU was suspended in DMSO/0.9% NaCl (1:4) and one dose (25 mg/kg) was intraperitoneally administered 24 h before euthanization. The dosage was chosen considering its safety and efficacy in downregulating hyaluronan production, as previously demonstrated [48,49,50].

Animal care and experimental protocols were approved by the Animal Care and Use Ethics Committee of the University of Insubria and by the Italian Ministry of Health (authorization number: n°415/2016-PR) and were performed in accordance with national and EU guidelines for the handling and use of experimental animals.

### 4.2. Intestinal IRI

Animals were anesthetized with thiopental sodium (50 mg/kg diluted 2% *w*/*v* in sterile isotonic saline and given i.p.) in a non-fasted state. One or two additional doses, consisting of 10% of the initial dose, were occasionally administered 20 min after the previous dose to maintain the anesthetic state. Intestinal IRI was obtained as previously described [51]. Briefly, after laparotomy a loop of the small intestine was exteriorized and a branch of the superior mesenteric artery (SMA) supplying the segment was temporarily occluded for 60 min with an atraumatic microvascular clamp. The upper and lower margins of the intestinal segment receiving blood from the clamped SMA branch were also clamped for 60 min. Animals were allowed to regain consciousness and were euthanized by decapitation 24 h after reperfusion, when major histopathological, immune, and neuromuscular functional changes at the intestinal level were observed [51]. Brains were immediately removed, and the hippocampus and striatum were rapidly dissected, frozen on dry ice, and stored at −80 °C for Western blot and qRT-PCR analysis. For immunofluorescence staining, the striatum and hippocampus were immediately fixed after collection.

### 4.3. Immunofluorescence Analysis

Immunofluorescence analysis was performed on paraffin-embedded hippocampus and striatum cross-sections obtained from all six experimental groups according to the method described in Bistoletti et al. (2019) [52]. In brief, after dissection, the hippocampus and striatum were postfixed at 4 °C in 4% paraformaldehyde in phosphate buffer (PBS composition in mM: 0.14 NaCl, 0.003 KCl, 0.015 Na_2_HPO_4_, 0.0015 KH_2_PO_4_, pH 7.4) for 4 h. Brain regions were treated for 72 h in sucrose 30% *w*/*v* in PBS and embedded in paraffin. Cross-sections (7 μm) were treated for 30 min with PBS containing 2% bovine serum albumin (BSA) before overnight incubation with the primary antibody (4 °C). After washing, incubation with an appropriate secondary antibody was performed for 60 min in a dark humid chamber. The details of the primary and secondary antibodies and related optimal dilution are reported in Table 1. PBS buffer used for washing steps and primary antibodies dilutions contained 2% bovine serum albumin (BSA). Control samples were incubated with 2% BSA in PBS. For the double immunolocalization experiments, sections were rehydrated and pre-incubated with a BSA blocking solution to minimize nonspecific binding, as previously described. Subsequently, samples were incubated with rabbit anti-IL6 and anti-S100*β* primary antibodies for 60 min, followed by a 45 min incubation with FITC- and Cy3-conjugated goat anti-rabbit secondary antibodies, respectively. Since the primary antibodies were generated in the same species, an alternative method was performed. Following the first staining cycle, sections were incubated with anti-rabbit IgG (Sigma) at a dilution of 1:500 for 2 h in order to block the possible binding of the secondary antibody used in the second staining cycle with the primary antibody applied during the initial stage [27]. Nuclei were then stained by incubating for 3 min with 4′,6-diamidino2-phenylindole (DAPI; 0.1 mg/mL in PBS, excitation 340 nm, emission 488 nm). Coverslips were mounted with Citifluor mounting medium and slides were finally examined with a Nikon fluorescence microscope. Images were combined with Adobe Photoshop (Adobe Photoshop version 5.5, 1999, Adobe Systems Inc, San Jose, CA, USA). Hyaluronan was assayed by using biotinylated hyaluronan binding protein as previously described [53].

### 4.4. RNA Isolation and Quantitative RT PCR

Total RNA was extracted from hippocampus and striatum samples with TRIzol (Invitrogen, Carlsbad, CA, USA) and treated with DNase I (DNase Free) to remove possible traces of contaminating DNA. cDNA was obtained by retrotranscribing **1** µg of total RNA using HiScript III 1st Strand cDNA synthesis Kit (+gDNA wiper, Vazyme, Red Maple Hi-tech Industry Park, Nanjing, China). Quantitative reverse transcription PCR (qRT-PCR) was performed on the QuantStudio 3 Real-Time PCR System (ThermoFisher Scientific, Carlsbad, CA, USA) with AceQ Universal SYBR qPCR Master Mix (Vazyme, Nanjing, China) following the manufacturer’s instructions. Primers were designed using Primer Express software (version 3.0.1 for Windows, 2020, Applied Biosystems, Milan, Italy) and used at a final concentration of 500 nM for each primer. Primers are reported in Table 2 and were designed to have a similar amplicon size and similar amplification efficiency as required for applying the 2^−ΔΔCt^ method to compare gene expression in the different groups subjected to intestinal IRI in the absence and presence of 4-MU treatment and in the CTR group treated with 4-MU with respect to CTR values [22]. *ACTβ* was used as a housekeeping gene. Experiments were performed at least five times for each experimental group.

### 4.5. Western Immunoblot Analysis

Hippocampus and striatum were used to analyze phospho-IκΒ (pIκBα), Toll-like receptor 4 (TLR4), and CD44 protein levels. Briefly, protein samples were boiled for 5 min at 95 °C in Laemmli sample buffer (Tris-HCl 300 mM, pH 6.8, glycerol 10%, SDS 2%, β-mercaptoethanol 0.04%) for protein denaturation, processed for electrophoretic separation and then blotting as described elsewhere [22]. Protein quantification was carried out with the Bradford method. Protein separation was carried out on 8% SDS-polyacrylamide gel electrophoresis (SDS-PAGE) and electroblotted to nitrocellulose membranes (Merck Millipore, Milan, Italy). Membranes were analyzed according to the protocol described in Bosi et al. [22], using ACTβ as a loading control. Dilutions and main features of primary and secondary antisera are reported in Table 1. Bands were visualized by chemiluminescence (LiteAblot, Euroclone SpA, Milan, Italy), acquired and quantified with Alliance Q9 Advanced System (version 18.16, 2024, Uvitec Ltd., Cambridge, UK). Experiments were performed at least five times for each experimental group.

### 4.6. Chemicals

All chemicals were obtained from Sigma–Aldrich (Milan, Italy), unless otherwise specified, and were of the highest commercially available analytical grade, with the lowest grade of 96%.

### 4.7. Statistical Analysis

All results are reported as mean ± standard error of the mean (SEM) of at least five experiments. Statistical significance was calculated using unpaired Student’s *t* test or by one-way ANOVA followed by Tukey’s post hoc test, when appropriate. Differences were considered statistically significant when *p* < 0.05. For statistical analysis the GraphPad Prism software was used (GraphPad Prism version 8.0.2 for Windows, 30 January 2019, GraphPad Software Inc, San Diego, CA, USA).

## Figures and Tables

**Figure 1 ijms-26-10064-f001:**
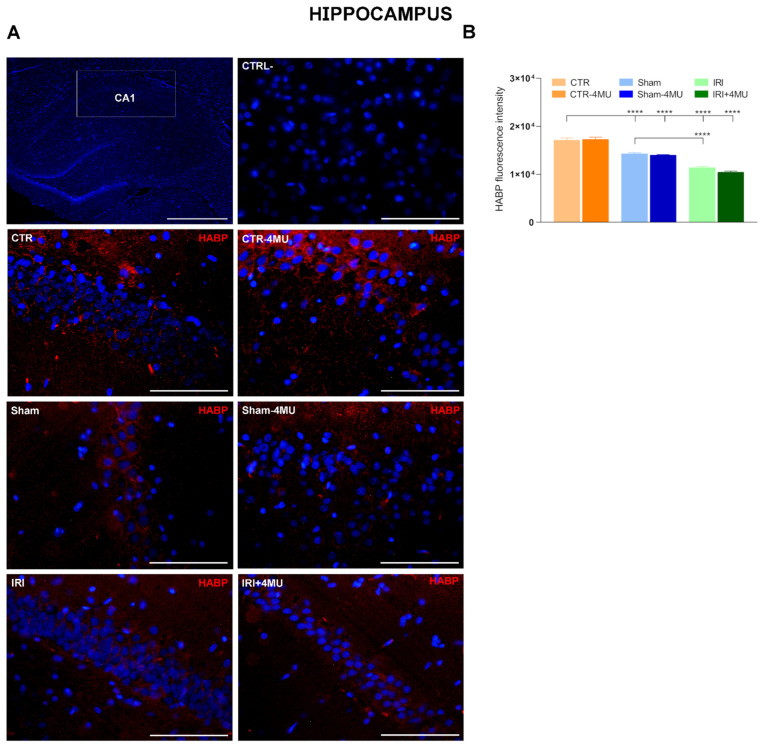
Hyaluronan immunofluorescence in the rat hippocampus after intestinal IRI: modulation by 4-MU. (**A**) Fluorescence microscopy images showing in red HABP immunostaining in the CA1 region of the hippocampus from the different experimental groups in the absence or presence of 4-MU treatment (25 mg/kg i.p.) as indicated on each panel and counterstained with DAPI (blue) (bars: 50 µm). The upper panel on the left represents a panoramic image of the hippocampus (bar: 500 µm), the upper panel on the right is a negative control (bar: 50 µm). (**B**) HABP fluorescent intensity obtained from the different experimental groups. N = 5 rat/group. Data are reported as mean ± SEM **** *p* < 0.0001, by one-way ANOVA with Tukey’s post hoc test.

**Figure 2 ijms-26-10064-f002:**
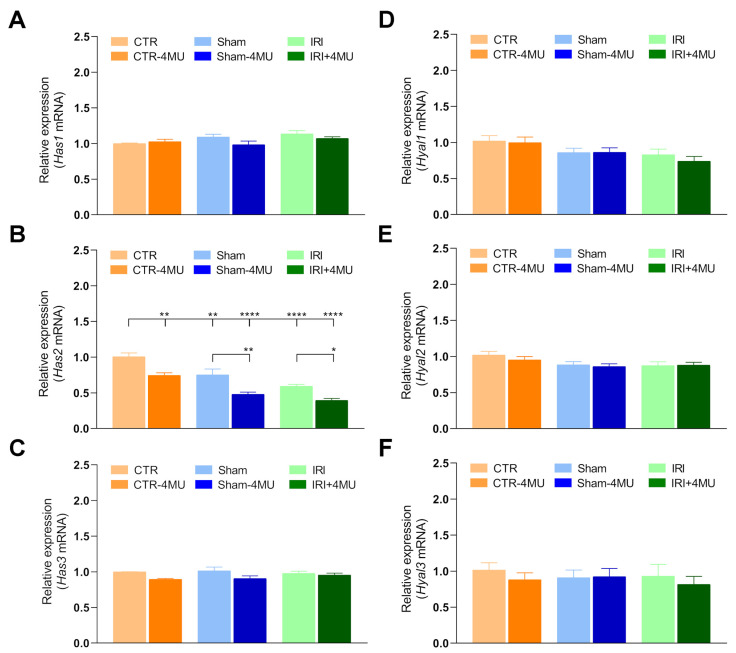
Hippocampal expression profiles of hyaluronan synthases and hyaluronidases after intestinal IRI. (**A**–**C**) *Has1*, *Has2*, *Has3* and (**D**–**F**) *Hyal1*, *Hyal2*, *Hyal3* mRNA levels obtained from the different experimental groups. Graphs show Hases and Hyales relative gene expression to *ACTβ*. N = 5 rat/group. Data are reported as mean ± SEM **** *p* < 0.0001, ** *p* < 0.01, * *p* < 0.05 by one-way ANOVA with Tukey’s post hoc test.

**Figure 3 ijms-26-10064-f003:**
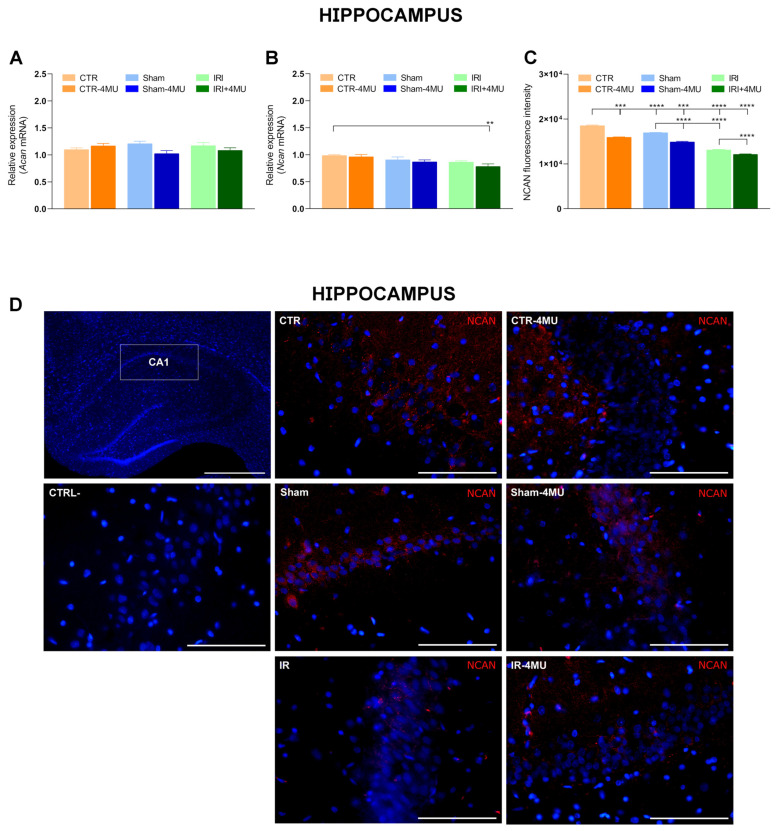
Influence of intestinal IRI on chondroitin sulfate proteoglycans, aggrecan (ACAN), and neurocan (NCAN) expression in the rat hippocampus after IRI and modulation by 4-MU. (**A**) *Acan* and (**B**) *Ncan* mRNA levels in the rat hippocampus obtained from the different experimental groups in the absence or presence of 4-MU treatment (25 mg/kg i.p.). Graphs show aggrecan and neurocan relative gene expression determined by calculating 2^−ΔΔCt^ and normalized to *ACTβ*. (**C**) NCAN fluorescent intensity obtained from the different experimental groups. (**D**) Fluorescence microscopy images showing in red NCAN immunostaining in the CA1 region of the hippocampus from the different experimental groups, as indicated on each panel and counterstained with DAPI (blue) (bars: 50 µm). The upper panel on the left represents a panoramic image (bar: 500 µm), while the lower panel on the left is a negative control (bar: 50 µm). N = 5 rat/group. Data are reported as mean ± SEM **** *p* < 0.0001, ** *p* < 0.01, *** *p* < 0.001, one-way ANOVA with Tukey’s post hoc test.

**Figure 4 ijms-26-10064-f004:**
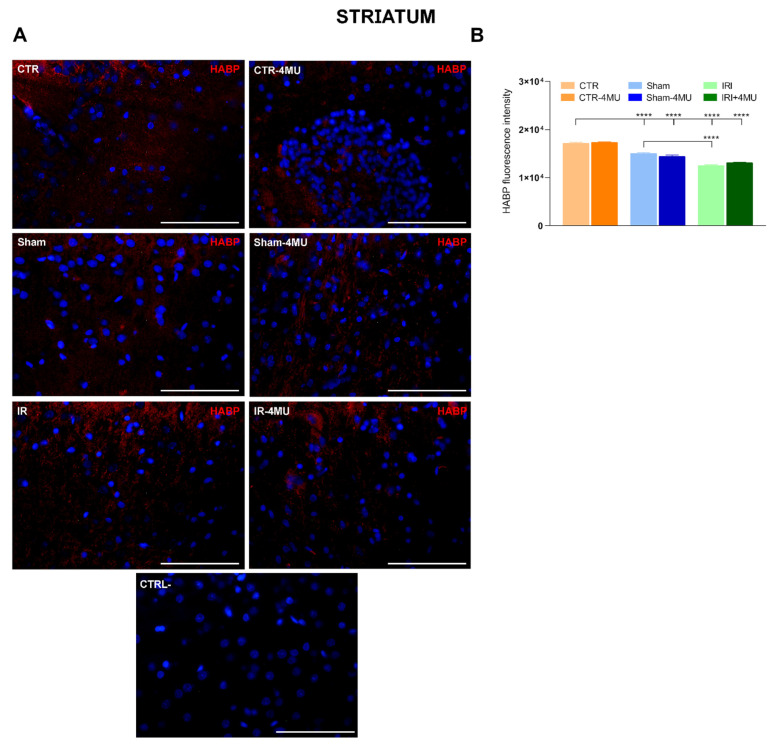
Hyaluronan immunofluorescence in the rat striatum after intestinal IRI: modulation by 4-MU. (**A**) Fluorescence microscopy images showing in red HABP immunostaining in the rat striatum from the different experimental groups in the absence or presence of 4-MU treatment (25 mg/kg i.p.) as indicated on each panel and counterstained with DAPI (blue) (bars: 50 µm). The lower panel represents a negative control (bars: 50 µm). (**B**) HABP fluorescent intensity obtained from the different experimental groups. N = 5 rat/group. Data are reported as mean ± SEM; **** *p* < 0.0001 by one-way ANOVA with Tukey’s post hoc test.

**Figure 5 ijms-26-10064-f005:**
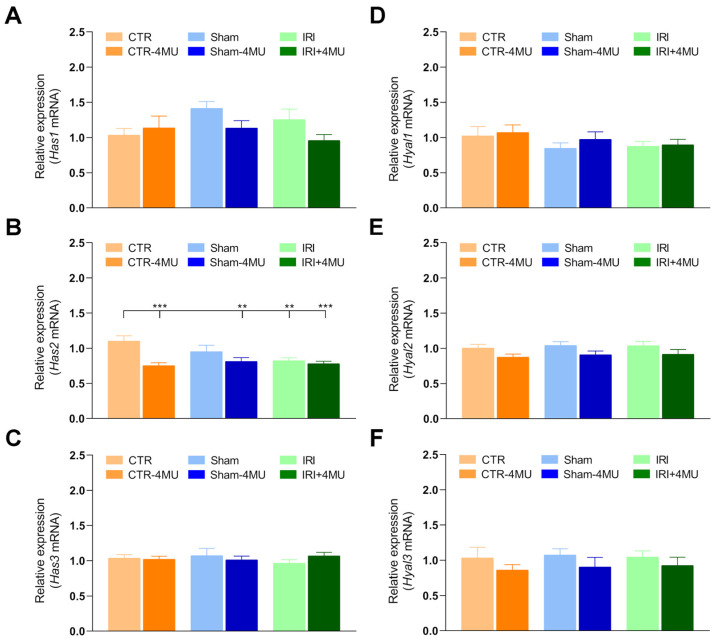
Striatal expression profiles of hyaluronan synthases and hyaluronidases after intestinal IRI. (**A**–**C**) *Has1*, *Has2*, *Has3* and (**D**–**F**) *Hyal1*, *Hyal2*, *Hyal3* mRNA levels obtained from the different experimental groups. Graphs show Hases and Hyales relative gene expression to *ACTβ*. N = 5 rat/group. Data are reported as mean ± SEM *** *p* < 0.001, ** *p* < 0.01, by one-way ANOVA with Tukey’s post hoc test.

**Figure 6 ijms-26-10064-f006:**
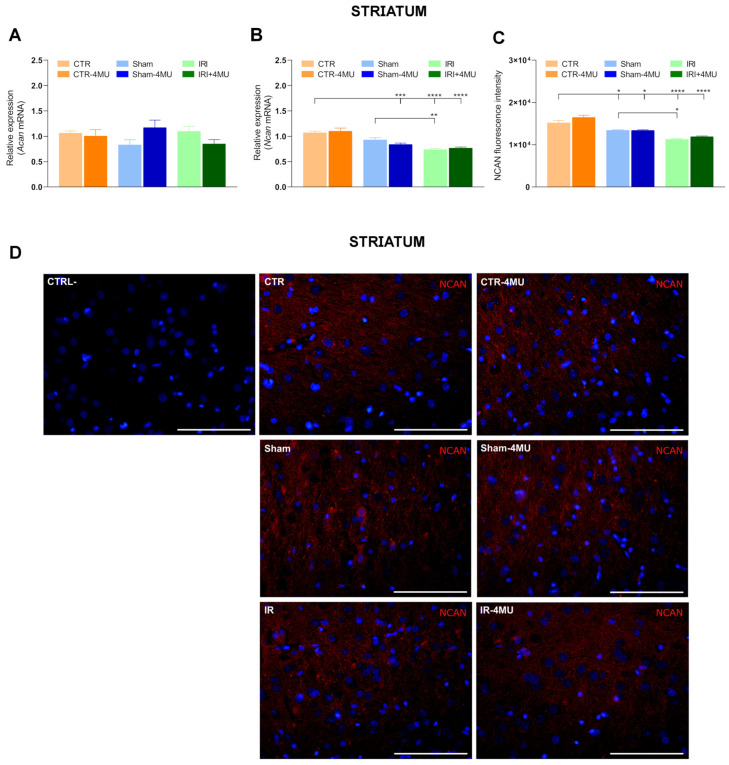
Influence of intestinal IRI on chondroitin sulfate proteoglycans, aggrecan (ACAN), and neurocan (NCAN) expression in the rat striatum after intestinal IRI and modulation by 4-MU. (**A**) *Acan* and (**B**) *Ncan* mRNA levels in the rat striatum obtained from the different experimental groups in the absence or presence of 4-MU treatment (25 mg/kg i.p.). Graphs show *Acan* and *Ncan* relative gene expression determined by calculating 2^−ΔΔCt^ and normalized to *ACTβ*. (**C**) Neurocan fluorescent intensity obtained from the different experimental groups. (**D**) Fluorescence microscopy images showing in red NCAN immunostaining in the rat striatum from the different experimental groups, as indicated on each panel and counterstained with DAPI (blue) (bars: 50 µm). The upper panel on the left represents a negative control (bars: 50 µm). N = 5 rat/group. Data are reported as mean ± SEM; **** *p* < 0.0001, *** *p* < 0.001, ** *p* < 0.01, * *p* < 0.05 one-way ANOVA with Tukey’s post hoc test.

**Figure 7 ijms-26-10064-f007:**
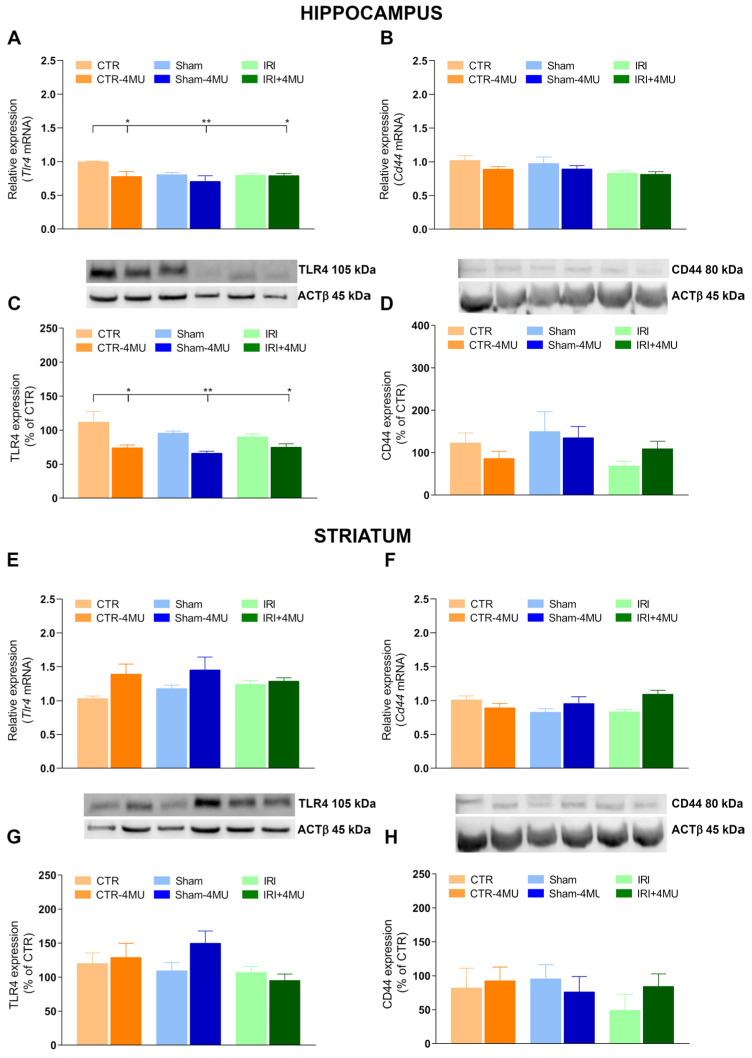
Effect of 4-MU on TLR4 and CD44 expressions in the rat brain after intestinal IRI. *Tlr4* and *Cd44* mRNA levels in the hippocampus (**A**,**B**) and striatum (**E**,**F**) obtained from the different experimental groups in the absence or presence of 4-MU treatment (25 mg/kg i.p.). Graphs show *Tlr4* and *Cd44* relative gene expression determined by calculating 2^−ΔΔCt^ normalized to *ACTβ*. TLR4 (**C,G**) and CD44 (**D**,**H**) protein levels in the hippocampus (**C**,**D**) and striatum (**G**,**H**) of the different experimental conditions. Blots representative of immunoreactive bands for TLR4, CD44 and ACTβ. Samples (80 µg) were electrophoresed in SDS-8% polyacrylamide gels. Numbers at the margins of the blots indicate relative molecular weights of the respective protein in kDa. Data are expressed as the optical density (O.D.) ratio of TLR4 and CD44 vs. ACTβ and normalized as a percentage relative to the CTR group. N = 5 rat/group. Data are reported as mean ± SEM; ** *p* < 0.01, * *p* < 0.05 by one-way ANOVA with Tukey’s post hoc test.

**Figure 8 ijms-26-10064-f008:**
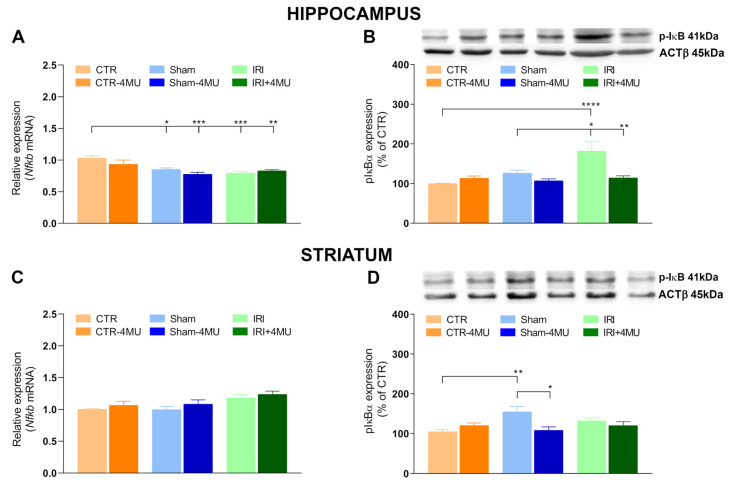
Effect of 4-MU on NF-*κ*B and pIκBα in the rat brain after intestinal IRI. (**A**,**C**) *Nfkb* mRNA levels in the (**A**) hippocampus and (**C**) striatum obtained from the different experimental groups in the absence or presence of 4-MU treatment (25 mg/kg i.p.). Graphs show *Nfkb* relative gene expression determined by calculating 2^−ΔΔCt^ values normalized to *ACTβ*. (**B**,**D**) pIκBα protein levels in the (**B**) hippocampus and (**D**) striatum in the different experimental conditions. Blots representative of immunoreactive bands for pIκBα and ACTβ. Samples (80 µg) were electrophoresed in SDS-8% polyacrylamide gels. Numbers at the margins of the blots indicate relative molecular weights of the respective protein in kDa. Data are expressed as the optical density (O.D.) ratio of pIκBα vs. ACTβ and normalized as a percentage relative to the CTR group. N = 5 rat/group. Data are reported as mean ± SEM; **** *p* < 0.0001, *** *p* < 0.001, ** *p* < 0.01, * *p* < 0.05 by one-way ANOVA with Tukey’s post hoc test.

**Figure 9 ijms-26-10064-f009:**
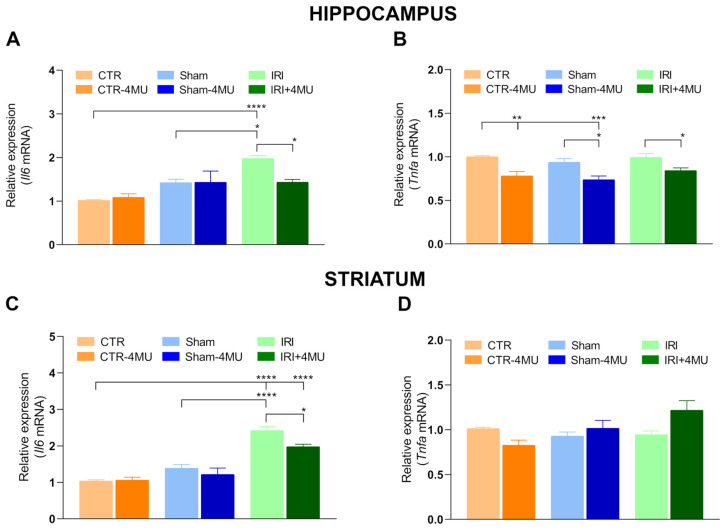
Effect of 4-MU on inflammatory cytokine mRNA levels in the rat hippocampus and striatum after intestinal IRI. (**A**–**D**) *Il6* and *Tnfa* mRNA levels in the hippocampus (**A**,**B**) and striatum (**C**,**D**) obtained from the different experimental groups in the absence or presence of 4-MU treatment (25 mg/kg i.p.). Graphs show *Il6* and *Tnfa* relative gene expression determined by calculating 2^−ΔΔCt^ and normalized to *ACTβ*. N = 5 rat/group. Data are reported as mean ± SEM **** *p* < 0.0001, *** *p* < 0.001, ** *p* < 0.01, * *p* < 0.05 by one-way ANOVA with Tukey’s post hoc test.

**Figure 10 ijms-26-10064-f010:**
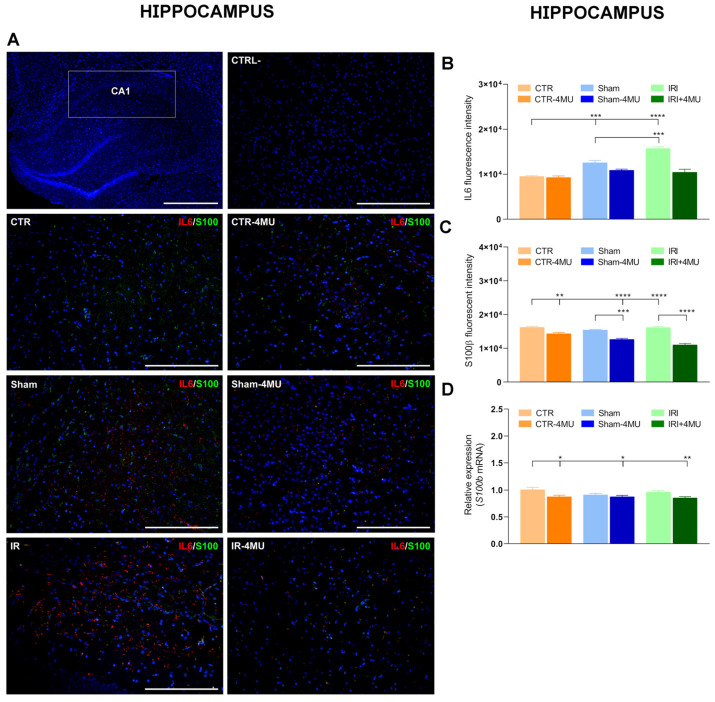
Influence of 4-MU on intestinal IRI-induced changes in IL6 and the microglial marker, S100β, in the hippocampus. (**A**) Fluorescence microscopy images showing green S100β and red IL6 immunostaining in the CA1 region of the rat hippocampus from the different experimental groups in the absence or presence of 4-MU treatment (25 mg/kg i.p.) as indicated on each panel and counterstained with DAPI (blue) (bars: 100 µm). The upper panel on the left represents a panoramic image (bars: 500 µm); the upper panel on the right is a negative control (bars: 100 µm). (**B**) IL6 and (**C**) S100β fluorescent intensity obtained from the different experimental groups. (**D**) *S100β* relative gene expression determined by calculating 2^−ΔΔCt^ and normalized to *ACTβ*. N = 5 rat/group. Data are reported as mean ± SEM **** *p* < 0.0001, *** *p* < 0.01, ** *p* < 0.01, * *p* < 0.05 by one-way ANOVA with Tukey’s post hoc test.

**Figure 11 ijms-26-10064-f011:**
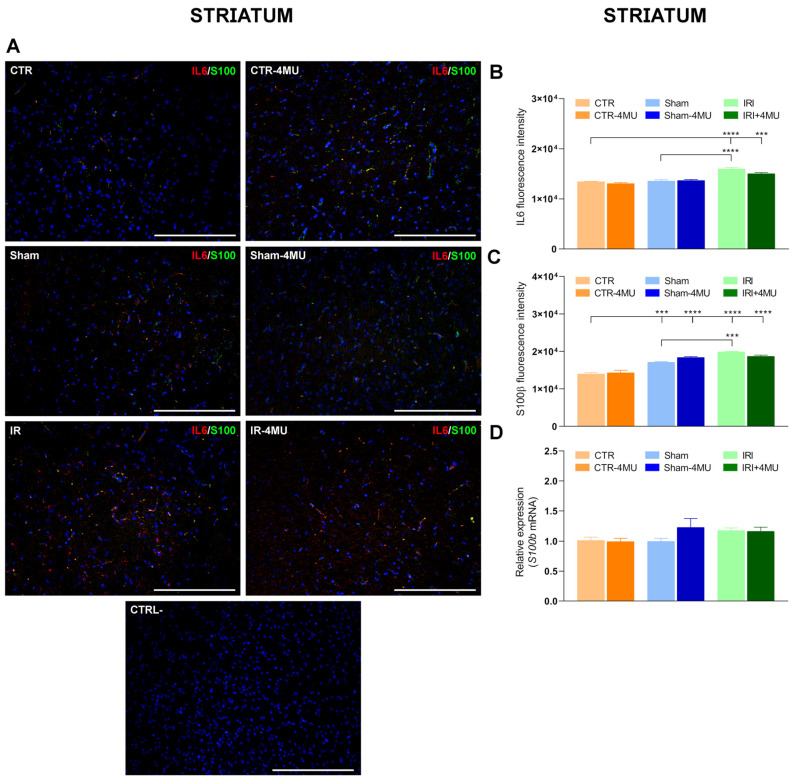
Influence of 4-MU on intestinal IRI-induced changes in IL6 and the microglial marker, S100β, in the striatum. (**A**) Fluorescence microscopy images showing green S100β and red IL6 immunostaining in the rat striatum from the different experimental groups in the absence or presence of 4-MU treatment (25 mg/kg i.p.) as indicated on each panel and counterstained with DAPI (blue) (bars: 100 µm). The lower panel represents a negative control (bars: 100 µm). (**B**) IL6 and (**C**) S100β fluorescent intensity obtained from the different experimental groups. (**D**) *S100β* relative gene expression to *ACTβ*. N = 5 rat/group. Data are reported as mean ± SEM **** *p* < 0.0001, *** *p* < 0.01 by one-way ANOVA with Tukey’s post hoc test.

**Table 1 ijms-26-10064-t001:** Primary and secondary antisera and their respective dilutions used for immunohistochemistry (HC), hyaluronan localization, and Western blot (WB) assay.

Primary Antibody	Host Species	Dilution (HC)	Dilution (WB)	Specificity	Source
HABP	-	1:100	-	Hyaluronan [53]	Hokudo (BC41)
S100*β*	Rabbit	1:100	-	S100*β* [54]	GeneTex (GTX129573)
IL6	Mouse	-	1:200	IL6 [55]	Santa Cruz (sc-57315)
IL6	Rabbit	1:150	-	IL6 [55]	Sigma Aldrich (I2143)
CD44	Rabbit		1:1000	CD44 [56]	Invitrogen (PA5-21419)
TLR4	Rabbit		1:1000	TLR4 [57]	ABclonal (A5258)
pIκBα	Mouse	-	1:400	pI*κ*Β−α [58]	Santa Cruz (sc-8404)
*ACT*β	Mouse	-	1:1000	Housekeeping [59]	Cell Signaling (#3700)
**Secondary Antibody** **& Streptavidin complex**					
Anti-rabbit Cy3-conjugated	Goat	1:200	-		Jackson (AB_2338006)
Anti-rabbit FITC-conjugated	Goat	1:200	-		Jackson (AB_2337972)
Streptavidin-Cy3	-	1:500	-		Amersham (PA43001)
Anti-rabbit IgG,HRP-linked	Donkey		1:2000		Amersham (NA934)
Anti-mouse IgG,HRP-linked	Horse	-	1:2000		Cell Signaling (#7076)

**Supplying companies:** Hokudo Co., Ltd., Sapporo, Japan; GeneTex, Alton Pkwy Irvine, CA, USA; Merk Life Science S.r.l., Milan, Italy; Santa Cruz Biotechnology Inc., Heidelberg Germany; Cell Signaling Technology, Danvers, MA, USA; Immunostar Inc., Hudson, WI, USA; Amersham, GE Healthcare, Buckinghamshire, UK; Jackson ImmunoResearch Cambridgeshire, UK; ABclonal, Wuhan, China.

**Table 2 ijms-26-10064-t002:** Sequence of primers used in the study for the qRT-PCR analysis.

Gene	Sequence
*ACTβ*	F 5′-CAGGGTGTGATGGTGGGTATGG-3′ R 5′-AGTTGGTGACAATGCCGTGTTC-3′
*Tnfa*	F 5′-CCCAGACCCTCACACTCAGAT-3′ R 5′-TTGTCCCTTGAAGAGAACCTG-3′
*Il6*	F 5′-GTGCAATGGCAATTCTGATTGTA-3′ R 5′-CTAGGGTTTCAGTATTGCTCTGA-3′
*Nfkb*	F 5′-CCTTTCTCTACTACCCCGAAATC-3′ R 5′-GAAGTTGGGCATGAGCTTCT-3′
*S100β*	F 5′-GGGTGACAAGCACAAGCTGAA-3′ R 5′-AGCGTCTCCATCACTTTGTCCA-3′
*Cd44*	F 5′-AAGACATCGATGCCTCAAAC-3′ R 5′-CTCCAGTAGGCTGTGAAGTG-3′
*Tlr4*	F 5′-TGAGATTGCTCAAACATGGC-3′ R 5′-CGAGGCTTTTCCATCCAATA-3′
*Has1*	F 5′-AGTATACCTCGCGCTCCAGA-3′ R 5′-ACCACAGGGCGTTGTATAGC-3′
*Has2*	F 5′-TTGGCCGGTCGTCTCAA-3′ R 5′-CGTCCTCCGCCTGTCTGT-3′
*Has3*	F 5′-CCTCATCGCCACAGTCATACAA-3′ R 5′-CCACCAGCTGCACCGTTAGT-3′
*Hyal1*	F 5′-GCCCATAATGCCCTACGTCCA-3′ R 5′-TGGCTTGGCATGACTCCTTG-3′
*Hyal2*	F 5′-GGAGCGGGCTTAGCTGGTA-3′ R 5′-GGGCTACAGGAAGTGTCACC-3′
*Hyal3*	F 5′-CACCAGATCCTCCACAACCT-3′ R 5′-GAGGCTGCCTGGTAGACTTG-3′
*Acan*	F 5′-TCCACATCAGAAGAGCCATAC-3′ R 5′-AGTCAAGGTCGCCAGAGG-3′
*Ncan*	F 5′-TTTCAGTCCACAGCGATCAG-3′ R 5′-AGGAGAGGGATACAGCAGCA-3′

## Data Availability

The data presented in this study are available on request from the corresponding author.

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
