# Peer review of "Intestinal Ischemia/Reperfusion Injury Influences Hyaluronan Homeostasis in the Rat Brain"

_ijms, 2025, doi:10.3390/ijms262010064_

Round 1
Reviewer 1 Report
Comments and Suggestions for Authors
Bosi et al present very interesting study about the effect intestinal ischemiaon influences hyaluronan homeostasis in the rat brain. The authors of this study indicated that intestinal IRI may contribute to the disruption of ECM homeostasis and induce hyaluronan -mediated enhancement of local proinflammatory signaling via mechanism mediated by TLR4 and NF-kB pathways. Although the authors invested a lot of work in this study. The analysis of both CD44 and TLR4 comes short in this study. Also, the expression of both seems to be influenced by the different treatments, compared to control. It is not clear, whether the stability of both TLR4 and CD44 is also influenced at the protein level? Also, the effects observed by the expression of TLR4 and CD44 in Hippocampus is exactly opposite to those in Striatum. Also, the analysis of p-IkB is not enough to determine the role of NF-KB and the correlation with the induced proinflammatory cytokines and to address the role of NF-kB in this study. This study need additional experiments to address the role TLR4, CD44-downsream pathways and the link to NF-kB pathway and the regulation of hyaluronan homeostasis in the rat brain.
Also, the presentation of the data is very confused. If this study really is qualified for the publication in the present form, I am very carious
Reviewer 2 Report
Comments and Suggestions for Authors
I reviewed the manuscript entitled Intestinal ischemia/reperfusion injury influences hyaluronan homeostasis in the rat brain.
I agree to accept this manuscript after major revision.
1) In the abstract section, the author used many abbreviations. In general, it is necessary to use abbreviations three or more times, otherwise too many abbreviations will confuse readers. Modify the abstract and the entire text according to this requirement.
2) 60 minutes should change to 60 min; 24 hours should change to 24 h. To use international units instead of words, please modify similar issues throughout the text.
3) TLR4-NFκB-pIκB, S100β, Greek letters need to be italicized (κ, β), check and revise the entire text.
4) Keywords: intestinal ischemia/reperfusion injury as the first keyword, it should be modified to Intestinal ischemia/reperfusion injury.
5) HA is a ubiquitous extracellular matrix (ECM) glycosaminoglycan formed… The first appearance of HA here should be changed to Hyaluronan (HA) is a ubiquitous...
6) 2.1. Intestinal IRI influences HA homeostasis and perineuronal net molecular components in the rat brain Each actual word in the secondary title needs to be capitalized. Please review and revise the entire text.
7) Figure 1. mean ± SEM, SEM should be written in full name. p< 0.0001, **p< 0.01, *p<0.05, when it comes to statistics, p should be italicized. Other Figures also have similar issues and need to be modified.
8) Na2HPO4, KH2PO4, These numbers should be subscripted.
9) I have read all the references and found some issues. All DOIs in the References need to be modified and a comma needs to be removed. For example, Ref 1, 10.1023/B:DDAS.0000042232.98927.91,. should change to 10.1023/B:DDAS.0000042232.98927.91. Ref 4, WEBSITE:WEBSITE:PERICLES;WGROUP:STRING:PUBLICATION. These contents need to be deleted. Ref 23, missing article number, I have found it for you, please add it. Article number: 17644. Ref 39, missing pages, I have found it for you, please add it. 4021-4044. Ref 42, missing article number, I have found it for you, please add it. Article number: e0152302. Refs 44, 46 and 48, the same issue exists, please make modifications.
10) Has statistical efficacy analysis been conducted to ensure that the sample size is sufficient to detect significant differences, with only 5 rats used in each group?
11) The SHAM group only exposed the arteries without occlusion, but it was not specified whether surgical trauma (such as vessel separation time) was simulated and whether it would affect the comparability of the results?
12) Why choose 24 h as the endpoint for reperfusion? Is there any preliminary data to support that this time point can fully reflect the dynamic changes of brain HA?
13) Has it been verified that administering medication only once 24 h before execution is sufficient to inhibit HA synthesis within this time window? Is there pharmacokinetic data to support it?
14) The article proposes that HA mediates inflammation, but although 4-MU reduces IL-6, it does not restore HA levels. Does this imply that HA plays an independent role in inflammation?
15) Is the different response of hippocampus and striatum to IRI related to differences in baseline HA metabolism or cell composition between the two? Additional discussion is needed.
16) The HA fragment mentioned in the article may promote inflammation, but the distribution of HA with different molecular weights has not been detected. Can it be verified by gel electrophoresis or ELISA?
17) Is it contradictory that TLR4 mRNA decreases but pIκBα increases? It is necessary to discuss whether TLR4 protein levels or downstream signals are activated.
18) The conclusion mentions that changes in HA may lead to cognitive impairment, but does not cite direct evidence. Is it necessary to supplement animal behavioral data?
19) 4-MU may have anti-inflammatory effects through non HA pathways (such as inhibiting STAT3), should these confounding factors be ruled out?
20) Will the differences in HA metabolism between rats and humans affect their clinical translational significance? It needs to be mentioned in the discussion.
21) This study investigated the role of hyaluronan (HA) in hippocampal and striatal alterations induced by intestinal ischemia-reperfusion injury (IRI) in rats. After 60 min of mesenteric ischemia and 24 h reperfusion, IRI animals showed significant downregulation of HA, neurocan, and HAS2 in both brain regions, with more pronounced effects in the hippocampus. These changes correlated with TLR4-NFκB-pIκB pathway disruption and increased IL-6 levels, which co-localized with microglial S100β. Treatment with the HA inhibitor 4-MU attenuated IL-6 elevation selectively in the hippocampus. The findings suggest that intestinal IRI disrupts ECM homeostasis, promoting HA-mediated neuroinflammation, particularly in the hippocampus, potentially contributing to cognitive deficits.
Round 2
Reviewer 1 Report
Comments and Suggestions for Authors
The manuscript is know imoroved and can be published in the present form
Reviewer 2 Report
Comments and Suggestions for Authors
The author has made the necessary modifications and explanations as per my request, therefore I agree to accept it in its current form.